# RNA Sensing of *Mycobacterium tuberculosis* and Its Impact on TB Vaccination Strategies

**DOI:** 10.3390/vaccines8010067

**Published:** 2020-02-04

**Authors:** Sanne Burkert, Ralf R. Schumann

**Affiliations:** 1Department of Infectious Diseases and Hematology, University Hospital of Ulm, 89081 Ulm, Germany; Sanne.Burkert@uniklinik-ulm.de; 2Department of Microbiology, Infectious Diseases and Immunology, Charité – University Medical Center Berlin, 12203 Berlin, Germany

**Keywords:** RNA, tuberculosis, vaccine, TLR8, NRLP3, RLRs, IL-12, IL-21, type I interferons

## Abstract

Tuberculosis (TB) is still an important global threat and although the causing organism has been discovered long ago, effective prevention strategies are lacking. *Mycobacterium tuberculosis* (MTB) is a unique pathogen with a complex host interaction. Understanding the immune responses upon infection with MTB is crucial for the development of new vaccination strategies and therapeutic targets for TB. Recently, it has been proposed that sensing bacterial nucleic acid in antigen-presenting cells via intracellular pattern recognition receptors (PRRs) is a central mechanism for initiating an effective host immune response. Here, we summarize key findings of the impact of mycobacterial RNA sensing for innate and adaptive host immunity after MTB infection, with emphasis on endosomal toll-like receptors (TLRs) and cytosolic sensors such as NLRP3 and RLRs, modulating T-cell differentiation through IL-12, IL-21, and type I interferons. Ultimately, these immunological pathways may impact immune memory and TB vaccine efficacy. The novel findings described here may change our current understanding of the host response to MTB and potentially impact clinical research, as well as future vaccination design. In this review, the current state of the art is summarized, and an outlook is given on how progress can be made.

## 1. Introduction

The tremendous increase in life expectancy of today compared to a century ago is in large part attributable to the unraveling of infectious diseases pathology followed by the development of successful prevention and treatment options against particular pathogens [1]. Both antibiotic therapy and vaccination concepts are based on profound knowledge about the biology of the pathogen and its interaction with the host. Several pathogens, i.e., those found to be slowly growing or composed of a specific cell wall differing in their induction of host immune responses still represent a particular challenge for both prevention and treatment. As a consequence, efficacious vaccines are still lacking for frequent and world-wide devastating diseases such as Malaria, HIV and TB. Although mycobacteria have been identified early as the cause of tuberculosis (TB), this disease is still a major global threat, leading to approximately 1.2 million deaths annually [2]. 

The basis for a long-lasting and protective immune response has been identified over the last century with the induction of a robust humoral response leading to the successful protection against a given pathogen. This “adaptive” immune response relies on the activation of antigen-presenting cells (APCs), which is directed by the innate immune system [3]. Particularly, the interaction of mycobacteria with innate and adaptive immune cells, however, differs from other bacteria and is still the focus of intense basic research. Over the last years, a large number of cell surface and intracellular receptors for bacterial compounds have been identified termed “pattern recognition receptors (PRRs)”, some of them recognizing “microbe-associated molecular patterns (MAMPs)” [4]. Two classes of bacterial compounds are recognized, one being cell wall compounds, the other nucleic acids, including RNA and DNA [5]. Here, we summarize recent findings on mycobacterial RNA recognition and its subsequent modulation of the host immune response through the induced cytokines IL-12, IL-21, and type I interferons, with its potential impact on TB vaccine development.

## 2. RNA Sensing and Activation of APCs

Evidence has been accumulated that bacterial RNA is an important “pathogen-associated molecular pattern (PAMP)” in the complex immune reaction initiated by infection with *Mycobacterium tuberculosis* (MTB) [6]. Through distinct pathomechanisms MTB evolved a way to survive intracellularly in macrophages and dendritic cells (DCs) in both phagosomes and the cytosol, using these APCs as their biological niche [7]. Thus, within these spaces, short-living mycobacterial RNA can activate an immune response and constitutes a sign of active bacterial infection (Figure 1); hence, it has recently been termed a “vita-PAMP” within these cells, playing a crucial role for activating a cascade of downstream immunological pathways [8].

MTB inhibits the formation of mature phagolysosomes. Antigens from infected phagosomes are secreted to the cytosol through the pore-forming 6 kDa early secretory antigenic target (ESAT-6) secretion system (ESX)-1. Phagosomal membrane disruption leads to translocation of the whole bacterium to the cytosol. Autophagy, an important process to build new phagolysosomes and eliminate mycobacteria, as well as to control excessive inflammasome-activation [9], is inhibited by mycobacterial virulence factors [10]. Thus, both endosomal (during the early inflammatory phase) and cytosolic receptors (during the later inflammatory phase) are able to detect RNA during MTB infection. PRRs involved in endosomal RNA recognition are toll-like receptors (TLRs) TLR3, -7, and -8 (Figure 1, pathway a). In the cytosol, RNA is recognized by the nucleotide-binding domain, leucine-rich-containing family, pyrin domain-containing-3 (NLRP3), the nod-like receptors (NRLs), and the rig-I-like receptors (RLRs) retinoic acid-inducible gene I (RIG-I), melanoma-differentiated gene 5 (MDA5) and laboratory of genetics and physiology 2 (LGP2), as well as oligoadenylate synthetases (OAS) and protein kinase R (PKR) (Figure 1, pathway b). Predominantly TLR3 and -8 have been linked to TB, as well as NLRP3, on which will be the focus in the following paragraphs.

### 2.1. Endosomal Mycobacterial RNA Sensing

#### 2.1.1. TLR8—The Most Prominent Phagosomal RNA Sensor 

TLR8 is expressed in macrophages and myeloid DCs (mDCs) and recognizes uridine-rich and short ssRNA making it a broad nucleoside sensor [11]. The gene for TLR8 is located at Xp22, which might explain the gender-specific differences frequently observed in infectious disease susceptibility. After infection with viable intracellular pathogens, interaction with the adapter protein MyD88 leads to NFκB-dependent expression of Interleukin (IL)-6, IL-12, IL-27, tumor necrosis factor alpha (TNFα) and interferon (IFN) γ, thus promoting a Th1 response, as well as interferon response factor (IRF)-7- and IRF5-triggered production of type I IFNs (Figure 1 pathway a). This pathway was shown to be of importance for multiple pathogens including RNA viruses [12] and different bacteria [13,14]. In human macrophages infected with *Mycobacterium avium*, a mycobacterium that cannot translocate into the cytosol due to lack of the ESX system, recognition, and degradation of intraphagosomal bacteria is largely dependent on their viability and myeloid differentiation primary response 88 (MyD88)-dependent TLR7/8 signaling [15]. TLR8 is generally thought to be dysfunctional in mice, but in the absence of TLR7 compensatory upregulation of TLR8 has been perceived [16]. Thus, it was possible to stimulate murine DCs with CLO75, a ligand claimed to be specific for TLR8, upon which they produced IL-12 [17]. Mice expressing human TLR8 show an increased frequency of autoimmune disorders [18], which supports the general importance of TLR8 in autoimmune diseases. Regarding TB, mice are known to be less susceptible and murine macrophages can resist BCG infection better than human ones [19]. TLR8-humanized mice permitted higher bacillary load in the lung [20], thus potentially impairing host immunity. On the other hand, knockdown of TLR8 in the human monocytic cell line THP-1 led to an increase in apoptosis upon BCG infection [21] and recent experiments with different MTB strains have linked higher levels of apoptosis of human DCs with less virulent strains [22].

Lately, functional studies have revealed a more exact understanding of TLR8-triggered immune responses: Stimulation of human peripheral blood mononuclear cells (PBMCs) with mycobacterial tRNA via TLR8 led to an early signal of IL-18, followed by the expression of IFNγ, IL12p70, and type I IFNs, the latter inducing a positive-feedback loop in natural killer (NK) cells to produce more IFNγ and thus promoting more production of IL12p70 [23] (Figure 1, pathway a). Moreover, bio-analytic tools confirmed that tRNA of mycobacterial culture induces a strong pattern of Th1 differentiation by endosomal RNA recognition, mainly dependent on TLR8. As RNA is only available during active infection, this might explain why there are multiple reports of a failure to induce active IL12p70 by dead bacteria, for example in a study investigating the IL23/IL17-axis in human infection with Mycobacterium avium [24]. For studies aiming at investigating the interaction of APCs and T-cells, the importance of viability of bacteria, i.e., RNA expression, thus could be crucial when trying to find in vitro models resembling in vivo conditions. Furthermore, it was demonstrated that neutrophils reacted upon TLR8 activation with the production of IL-6 and TNFα, as well as IL-23, thus promoting TH17 differentiation [25]. 

TLR8 has also been linked to mycobacterial infection by genetic analysis of single nucleotide polymorphisms (SNPs) of TLR8 and its correlation with disease susceptibility. The SNP TLR8 A1G (rs3764880: M > V) changes the start codon position from the first exon to the second, leading to an N-truncated variant depleted of three amino acids, which was reported to be hypermorphic [26]. Next to other intracellular infections such as HIV and HCV [27,28], this SNP was linked to TB, as the truncated version of TLR8 +1G has been found to protect against active pulmonary TB in Indonesian, Russian, and Moldavian males [29,30], in Turkish children [31], and in two Indian cohorts for both pulmonary TB (PTB) and extra-pulmonary TB (EPTB) [8,32]. In a Han Chinese cohort, another SNP at position −129 in the promoter region (rs3764879: C > G) has been reported to be in complete linkage disequilibrium with TLR8 A1G and, similarly, the allele with enhanced promoter activity exhibited an enhanced reactivity to RNA in the HEK-cell-model, as well as inactivation of MTB in PBMCs [33], thus linking a stronger activation of the TLR8 pathway with protection against TB. In contrast to that, two studies, from Pakistan [34] and among Han Chinese [35], have found that the G-Allele of TLR8 A1G is linked to susceptibility for TB.

#### 2.1.2. TLR7—ssRNA Sensor in pDCs

TLR7, the other receptor recognizing ssRNA/nucleosides in the endosome, is primarily expressed in plasmacytoid dendritic cells (pDCs) [36] and its stimulation leads to a type I IFN response. As TLR8 in mice is nonfunctional, murine TLR7 activation is often seen when in humans TLR8 is found to be responsive. It is known that human TLR8 activation inhibits expression of TLR7 and TLR9 in human embryonal kidney (HEK) cells and in human monocytes, though in tissue macrophages the potential of synergy has been proposed [37]. Regarding TB, murine macrophages showed an upregulation of TLR7 after stimulation with mycobacterial RNA [38]. Furthermore, the authors showed that treating MTB-infected macrophages with ssRNA resulted in higher viability through enhanced autophagy. Regarding SNP analysis for TLR7, a mutation at IVS2-151 (rs179009) has been found to be linked to TLR8-129, thus forming a haplotype that is linked to TB disease, but there was no evidence for an independent significance for TLR7-151 [37].

#### 2.1.3. TLR3—Recognizing dsRNA

TLR3, found in all innate immune cells except pDCs and neutrophils, recognizes dsRNA and its activation leads to a TIR-domain-containing adapter-inducing interferon-β (TRIF)-dependent manner of type I IFN-production and induction of pro-inflammatory cytokines via NFκB, as well as to cross-priming of CD8+ T-cells [39]. Upregulation of TLR3 in epithelial lung cells was shown after infection with different strains of MTB [40]. Upon infection with MTB, TLR3 knockout mice showed diminished IL-10 levels and increased Il-12 production, higher Th1-cell counts in the spleen and a lower mycobacterial burden [41]; thus, hinting toward a detrimental effect of TLR3 activation. Another role for TLR3 as a target of mycobacterial virulence factors was recently suggested by mice studies, which revealed that MTB inhibits the activation of TLR3 via c-ABL-dependent regulation of bone morphogenesis protein (BMP) [42]. This is an interesting finding, as inhibitors such as imatinib targeting ABL are in broad clinical use and have been shown to potentially act synergistically with current first-line tuberculostatic drugs [43].

Another possible approach for further research on endosomal RNA-sensing of MTB could be the receptor for advanced glycation end products (RAGE), which has been found to enhance, among others, endosomal RNA transport of extracellular RNA [44]. Human DCs and monocytes during active TB exhibited an increase in RAGE expression on cell membranes [45], and among a Brazilian TB cohort, levels of soluble RAGE were higher than in healthy controls [46].

### 2.2. Cytosolic Mycobacterial RNA Sensing

Two to three days after infection, MTB will translocate from the phagolysosome to the cytosol upon liposomal membrane disruption mediated through a conformational change of ESX in an acidified environment [47,48]. This promotes MHC-I presentation, as well as activation of cytosolic PRRs such as the inflammasome, a cytosolic multiprotein complex whose activation leads to the activation of caspase-1, which promotes processing and release of IL-1β and IL-18, as well as the induction of pyroptosis (Figure 1, pathway b).

#### 2.2.1. NLRP3—The Cytosolic Multisensor

NLRP3, a protein of the cytosolic inflammasome complex, is activated by a wide range of PAMPS and danger-associated molecular patterns (DAMPs), such as crystal phagocytosis, radical oxygen species (ROS), an influx of calcium, potassium, ATP or poration of the lysosome, as well as mycobacterial dsRNA [49]. In the mouse model, after activation of NLRP3 by prokaryotic mRNA in monocytes/macrophages, activation of TRIF and IRF3 resulted in production of IFNβ, which hierarchically leads to a caspase-1-dependent production of IL-1β [50,51]. Additionally, NLRP3 and consequent IL-1β activation have been shown after ESAT-dependent release of Cathepsin B into the cytosol in mice [52]. 

For TB evidence for both, a protective role of NLRP3 and IL-1β, as well as a pathogenic immune reaction can be found: Mice defective of IL-1β are more susceptible for TB [53] and human macrophages isolated from three individuals with combined gain-of-function-mutations of NLRP3 (rs35829419, M299V—no assigned rs number yet) and Card8 (rs2043211) in vitro allow a decreased mycobacterial growth and higher levels of IL-1β [54]. Further, SNP analysis revealed that a mutation leading to higher cytosolic activity of NLRP3 (rs10754558) protects from PTB in a Brazilian cohort [55], which similarly has been linked to HIV and HTVL-1 infection. In contrast, a recent investigation of these SNPs in a large Ethiopian cohort has found some evidence for an association with poor clinical outcome of PTB [56] and it has also been associated with early mortality in HIV/TB co-infection in Botswana. This may possibly be attributed to higher levels of IL-18 and IL-10 [57]. In line with this, excessive IL-1β has been found to promote a detrimental immune response in chronic disease [58].

#### 2.2.2. Other Cytosolic Sensors

Lately, there has been accumulating evidence for the importance of other cytosolic RNA sensors in TB immunity, particularly the RLRs. RIG-1 and MDA5, known sensors of ssRNA/short dsRNA with a 5′-triphosphate terminus and long dsRNA, respectively, lead to an induction of type I IFNs. These have been linked to susceptibility to several nonviral, intracellular pathogens such as Listeria monocytogenes, Legionella pneumophila, Plasmodium falciparum, and the transiently intracellular Helicobacter pylori. Regarding TB, it could be shown that expression of RIG-I, MDA5, and laboratory of genetics and physiology 2 (LGP2), the third known RLR, was upregulated in bone-marrow-derived macrophages (BMDM) from mice upon stimulation with mycobacteria [59]. This induction was observed only upon stimulation with live but not dead bacteria. Furthermore, it was shown that activation of RIG-I and MDA5 inhibited mycobacterial growth in different cell lines [60]. There is also evidence for a synergy of the DNA-recognizing cyclic GMP-AMP synthase (cGAS)/stimulator of interferon genes (STING)-axis with the MDA5-IRF7 pathway, activated by cytosolic RNA secreted from the endosome leading to IFNβ expression [61]. LGP2 has a unique status, as it can activate type I IFN responses by interacting with MDA5, but also inhibit RIG-I and MDA5 triggered expression of IRF3 and NFκB by interaction with TNF receptor associated factor (TRAF) [62]. Similarly, 2′-5′-oligoadenylate synthases (OAS)-like protein (OASL), an interferon-induced antiviral protein, has been gaining attention because of its bi-functional role: With ssRNA from RNA viruses it triggers a type I IFN-response through RIG-I, and with DNA viruses following dsDNA sensing it inhibits type I IFN response through deactivation of cGAS/STING [63]. OASL has been shown to play this role in human infection with Mycobacterium leprae, a mycobacterium also containing the ESX-1 secretion system that triggers a strong type I IFN response through cytosolic nucleic acid recognition [64]. Furthermore, OASL blocked autophagy and thus supported cytosolic persistence of mycobacteria. For MTB, no associations for OAS/OASL have been found yet, but its linkage to type I IFNs and autophagy makes it plausible and likely.

The role of PKR, for which it was shown that sensing mycobacterial dsRNA leads to inhibition of translation, apoptosis, and ultimately decreased mycobacterial growth, is somewhat controversial as it has been recently questioned by retrospective investigation [65]: The observed difference between assumed KO mice in the PKR domain was attributed to different mice strains rather than to the impact of PKR itself. Nevertheless, recent cell line experiments could demonstrate that PKR activation by mycobacterial RNA leads to growth restriction in human cells [60].

In summary, cytosolic RNA sensing in TB immunity is a currently expanding research field with mounting evidence for its significance for pathogenesis. This may be of importance with regard to new therapeutic options, since for example the drug Nitazoxanide seems to activate this pathway leading to mycobacterial growth inhibition [60].

### 2.3. Synergy of Different PRRs

In a humanized mouse model, synergistic effects regarding cytokine production such as type I IFNs and IL-12 of mDCs have been described after simultaneous stimulation of TLR3 and -8 [66]. It was even shown that a single synthetic ligand of TLR8 was not sufficient to induce a strong IL-12p70 response, but activation of both TLR3 and -8 was needed to synergistically lead to a strong induction of IL-12p70 and IFNγ [23]. This indicates that a co-stimulation of at least two TLRs is needed to build a robust IL12p70 induction. Similarly, significantly enhanced DC maturation and IFNγ production of NK cells has been found after co-stimulation with TLR8 and either TLR3 or -4 [67]. Using Poly(I:C)-encapsulating nanoparticles as a TLR3 agonist after BCG infection or specific TLR2 activation led to synergistic production of nitric oxide (NO) in mural BMDM [68]. This was largely dependent on TRIF and completely dependent on the capability to produce type I IFNs and on MyD88, proposedly through TLR2. It was also shown that co-stimulation of TLR2 and -8 inhibits IFNβ production by suppression of IRF5 [69]. This study proposed that activation of both TLRs shift T-cell differentiation from Th1 to Th17 through higher levels of IL-23 at the cost of IL-12p70 [69].

Moreover, synergy with cytosolic RNA sensing was found, as upon stimulation of human monocytes with *Methanosphaera stadtmanae*, a prokaryotic archaeon abundant in the human gut, NLRP3 activation leading to production of IL-1β and pyroptosis was observed, which was completely dependent on TLR8 activation [70]. Similar results have been found in studies with human PBMCs, THP-1 and in vivo in a *Cynomolgus* monkey model: Stimulation with the TLR8-agonist Motolimod led to activation of NK cells by IFNγ, as well as activation of NLRP3 and the inflammasome. This resulted in the release of IL-1β and IL-18 in a caspase-1-dependent manner, the latter strongly augmenting the NK driven production of IFNγ [71]. Interestingly, also for HIV it has been shown that TLR8 activation by RNA is sufficient and necessary to induce pro-IL-1β, which then is activated upon NLRP3 stimulation [72]. Regarding the question how TLR8 activates NLRP3, reactive oxygen species (ROS) after TLR8 activation in the endosome seem to play a role: For human neutrophils, it was shown that TLR8 activation leads to production of ROS by NADPH oxidase 2 (NOX2) activation [73] (Figure 1, pathway a). In support of this, activation of the inflammasome by TLR8 in both studies was perceived to be dependent on Cathepsin-B, known to trigger NLRP3 activation after its release to the cytosol upon permeabilization of the lysosome [74]. Activation of the inflammasome by Gram-positive bacteria through NRLP3 regulates NOX2 by caspase-1 within the phagosome to promote early acidification, thus providing a negative-feedback loop [75] (Figure 1, pathway c). Of note, MTB inhibits these pathways by different mechanisms and thus avoids production of ROS and acidification of the phagosomes.

In summary, it is increasingly recognized that RNA sensing, both in the endosome and in the cytosol, plays a significant role for the host’s defense to MTB, and interaction of the involved receptors seems to be crucial. At the end of the cascade of mycobacterial RNA sensing, there is release of the key cytokines IL-12p70, IL-21, type I IFNs, and IL-1β from APCs. Furthermore, IFNγ is released from NK cells to further orchestrate both T-cell differentiation and activation of the innate immune system.

## 3. Downstream Pathways and T-Cell Activation after Mycobacterial RNA Sensing

Activated APCs, after having sensed RNA, migrate to the lymph node and present antigens via major histocompatibility complex (MHC)-II molecules to CD4+ T-cells and via MHC-I to CD8+ T- cells, as well as to γδ T-cells, and initiate further differentiation through the different cytokines induced by RNA sensing. Specific T-cells will return to the lung to exercise their function as an adaptive immune response. Traditionally, CD4+ T-cells are believed to be the most important compartment for mycobacterial infection control. In patients with active TB, blood CD4+ T-cell counts are generally diminished and slowly increase with anti-tuberculosis treatment [76]. Furthermore, patients with low CD4+ counts are known to be more susceptible to mycobacterial infection. The fact that mice are generally less susceptible to TB could be explained, among other causes, by a faster acquisition of effector CD4+ T-cells in the lung [77]. Recently, it was shown that infection of human DCs with different MTB-strains varied in their potency of activating an effective CD4+ cell response since less virulent strains triggered a strong CD4+ cell activation that contained the infection. In contrast, hypervirulent strains failed to induce CD4+ cell activation [22]. This was associated with a high and low rate of apoptosis, respectively. Generally, the CD4+ T-cell compartment can be further differentiated into: T-helper (Th)-1 cells, Th2 cells, Th17 cells, follicular helper T (TfH) cells, and regulatory T-cells (Tregs). Due to the intracellularity of MTB within macrophages and DCs, the provoked immune response primarily triggers the Th1 cell compartment and its effector cytokines IFNγ and TNFα, which are also the major cytokines induced by RNA sensing (Figure 2). Another source of these cytokines are cytotoxic CD8+ T-cells, as well as NK cells, further promoting macrophage activation and eventual clearance of infected macrophages.

### 3.1. IL-12—The Key Cytokine for a Th1/17 Response

As shown above, IL-12 is a major cytokine secreted by RNA-activated APCs and the key factor to generate a Th1 response, thus playing a pivotal role in the pathogenesis of TB (Figure 2, pathway a). The biological most active form is IL-12p70, a heterodimer of the p35 and the limiting p40 subunit. It is primarily produced by DCs, macrophages, and neutrophils. From mice experiments we know that knocking out IL-12p40 leads to the inability of controlling mycobacterial growth [78] attributed to less DC migration to the draining lymph node and subsequently less antigen-presentation and lower levels of activated CD4+ T-cells. Clinically it is known that pharmaceutical inhibition of TNFα in patients can lead to reactivation of latent TB. Evidence of the outstanding importance of Th1 cell-mediated immune response has also been accumulated by studies from primary immune deficiencies, namely from patients with a Mendelian susceptibility to mycobacterial diseases (MSMD). These patients usually suffer from BCG disease and infection from atypical mycobacteria in early childhood. Those genes already identified for causing MSMD all rank around an impaired IL-12- and IFNγ-dependent pathway of APC activation and CD4+ T-cell differentiation [79]. Along with the pro-inflammatory effect of an activation of Th1 cells, IL-12 also induces IL-10 and thus simultaneously triggers containment of the immune response; hence, murine DCs activated with the TLR7/8 agonist CLO75 induce Tregs to express the IL-12 receptor subunit beta 2 (IL-12RB2) via MHC-II molecules, which inhibit the Th1 immune response by consuming IL-12 [17] (Figure 2, pathway b).

The IL-12p40 subunit is also part of the IL-23 receptor and thus leads to formation of Th17 cells, which is why both T-cells are often summarized as the Th1/Th17 compartment. Th17 is also driven by IL-21, hence these cells are linked to both key cytokines induced by RNA recognition [80] (Figure 2, pathway c). Stimulation of TLR-8 by ssRNA of murine THP-1 cells led to a cytokine milieu favoring Th17 induction [81]. Th17 cells in TB are reviewed elsewhere in detail, but generally they have been linked to an initial protective host immune response towards MTB through neutrophilic attraction and activation by their main cytokine IL-17. This is promoting the formation and maturation of granulomas [82]. On the other hand, Th17 cells promote dissemination of MTB [83]. In active TB, the IL-17/Th17 compartment is controversial: Th17 cells have been found to be increased in blood of PTB patients [84], as well as decreased in active PTB as compared to healthy controls with an even greater reduction in EPTB [24]. Similarly, IL-17 levels have both been found to correlate positively [85] and negatively [86] with TB disease. Mice deficient in IL-17 had significantly decreased levels of IFNγ after BCG infection [87]. Regarding genetic studies, the hypermorphic A-Allele of a SNP of IL17A- 197A conveyed protection in a cohort of active PTB [88], although a recent meta-analysis revealed significance only in Latin America [89]. In summary, Th17 cells seem to play an important role in initial mycobacterial infection, but its overall impact is not clear.

It has also been shown that IL-12 leads to an expansion of a subtype of the ‘unconventional’ γδ T-cells derived from humans that secrete Th1 and cytotoxic cytokines upon stimulation with (E)-4-hydroxy-3-methyl- but-2-enyl pyrophosphate (HMBPP) [90]. Interestingly, immunization with respiratory listeria leading to an expansion of HMBPP-specific γδ T-cells was shown to be beneficial for subsequent containment of a pulmonary challenge of MTB in nonhuman primates [91].

### 3.2. IL-21 and the Significance of TfH Cells in TB

Next to the IL-12/TNFα/IFNγ-axis, IL-21 is the central cytokine induced by RNA recognition, which is increasingly recognized to play a role in mycobacterial pathogenesis. It has been shown, that RNA sensing is the crucial “vita-PAMP” inducing TfH differentiation dependent on the endosomal TLR8-IL-12-axis in humans [8] (Figure 2, pathway d). In mice, this is fundamentally different as it depends on the cytosolic pathway orchestrated by IL-1β and IL-18 after NLRP3 activation [51]. It was found that mice deficient in IL-21 have a higher bacterial burden in the lung after 16 weeks and ultimately decease earlier [92]. In line with this, knockout mice for the IL-21-receptor (IL-21R) had increased mortality after MTB infection displaying less Ag-specific CD4+ and CD8+ T-cells infiltrating the lung. Furthermore, lower levels of IFNγ, IL-17, IL-12, IL-1β, and IL-27, as well as less cytotoxic molecules such as perforin and granzyme B, but higher levels of transforming growth factor (TGF)-β and IL-10 were observed [93]. Thus, IL-21, the hallmark cytokine for human TfH cells, can be assumed to have a major role in TB immunity. Mice deficient for BCL-6, a major transcription factor necessary for TfH differentiation, showed dramatically reduced levels of MTB-specific CD4+ T-cell responses [94]. The main function of TfH cells is to promote Ig secretion from B-cells, to induce their switch into plasma cells [95], and the formation of memory B cells. Thus, it is indicated that importance in TB immunity seems to question the dogma of a negligible role of humoral pathways involved. Several protective antibodies in mice and humans against MTB have been found [96,97], and although B cell-mediated humoral response and their interaction with other types of cells might not be the central part of host defense against MTB, it is long known that they do contribute [98,99]. Hence, their role is currently under reviewed discussion [100]. Mice experiments suggest that B cells are important to contain neutrophilia in early infection through IL-17 so that DCs can migrate to the lymph node and fulfill their function as APCs [101].

IL-21 furthermore acts significantly on cytotoxic lymphocytes (Figure 2, pathway e). The main induction signal for CD8+ cytotoxic T-cells in TB in the draining lymph node of mice is IL-12, but in the lung it was also dependent on IL-21, and type I IFNs [92]. In viral infections, mice experiments have shown that IL-21 is necessary in chronic infection to maintain a CD8+ T-cell response [102]. It was suggested that IL-21-deficient mice might be more susceptible to chronic TB infection through less potent CD8+ T-cells. This was postulated to be caused by IL-21 being important for both, CD8+ T-cell priming, as well as inhibiting the expression of the exhaustion markers programmed death 1 (PD-1) and mucin domain-containing protein 3 (TIM-3) [92]. However, a different set of mice experiments could not confirm the link of IL-21 preventing PD-1 expression, but did show that IL-21 played a role in CD8+ T-cell differentiation and proliferation upon BCG challenge [103]. IL-21 has been found to promote NK activation in humans, leading to enhanced production of IFNγ, perforin, granzyme B, and granulysin. This enhanced lysis of infected macrophages and monocytes in an in vitro MTB model [104]. After BCG vaccination, the formation of NK-like memory cells, which could convey protection from MTB infection, was observed to be dependent on IL-21 in mice [105], thus underlining the major relevance for IL-21 in NK cells for both acute effector function, as well as memory cells. In summary, IL-21 shifts the balance of Th1/17 vs. Tregs-cells to the Th1-side and leads to activation and lysis of alveolar macrophages and monocytes, as well as promotes cytotoxic T-cell immune response and their analogues.

### 3.3. Type I Interferons and Their Potential Rediscovered Importance in Early TB Immunity

Next to IL-12 and IL-21, type I IFNs are induced by mycobacterial RNA sensing. The role of type I IFNs in TB pathogenesis is subject to current discussion. Generally, they were held to play a detrimental role regarding the control of mycobacterial growth by inhibiting IL-12 and TNFα production and responsiveness to IFNγ, proposedly through IL-10 [106]. Similarly, it has been shown that an impaired type I IFN production due to C-type II lectin dendritic cell immunoreceptor (DCIR) deficiency leads to higher IL-12p40 levels and Th1 response in mice resulting in better control of mycobacterial growth [107]. This confirms the crucial role of IL-12 triggered Th1 response in mycobacterial infection, as well as the negative impact of type I IFNs for infection control. On the other hand, there are results indicating that type I IFNs also have a necessary part in infection control, for example in the absence of IFNγ in murine macrophages [108]. In human blood, it has been shown lately that the compartment of pDCs, mDCs2, and their capability to produce IFNα has been impaired in patients with active PTB, but not latent [109]. This may promote an important role of IFNα in early infection by DC maturation and antigen-presentation to CD8+ cells. In line with this, it has been shown in vitro that upon RNA stimulation the production of IFNγ from NK cells in human PBMCs vastly depends on a prior production of type I IFN in the sense of a feed-forward loop [23] (Figure 1, pathway c). Moreover, in studies with pleural fluid cells from patients with active TB it was shown that in addition to spontaneous expression of IFNβ by CD14+ cells, stimulation with BCG in combination with IFNβ leads to an enhanced IFNγ and decreased IL-17 production [110]. Neutrophilic production of IL-6 upon stimulation of TLR8 could also be significantly enhanced with IFNα [97]. The synergistic effect of enhancing pro-inflammatory NFκB expression in murine BMDM via TLR3 was also found to be largely dependent on type I IFNs [68]. Thus, there is evidence to review the role of type I IFNs in TB particularly during early infection as an auto-/paracrine enhancer of innate immunity. Furthermore, one might be tempted to speculate, that, as mice lack TLR8 and thus provide a stronger TLR7 response with potentially stronger induction of type I IFNs, this could be one of the reasons why mice are less susceptible to TB.

## 4. The Impact of RNA Sensing on Vaccination

The only currently licensed vaccine for TB is an attenuated strain of *Mycobacterium bovis*, BCG, which gives protection mainly from disseminated disease in childhood with differently reported efficacy but fails to implement satisfying protection of pulmonary disease in adolescence, when we see a peak of prevalence. Still, BCG is the widest used vaccine in the world. Advantages in the development of new TB vaccines have recently been extensively reviewed [111,112,113,114], therefore in this review, we focus on immunological mechanisms in vaccine-strategies with regard to pathways involving RNA sensing (Figure 3).

Generally, attenuated vaccines with viable pathogens are known to be more efficient in implementing a strong memory immune response as compared to inactivated ones, but they convey a higher risk for immunosuppressed individuals of an infection. Especially with increasing co-infection of TB and HIV, it is highly desirable to better understand the cause of higher protection by viable bacteria in comparison to dead antigens in order to design a better dead vaccine with a higher safety profile. Thus, mimicking live attenuated vaccines by activation of RNA sensing pathways might overcome the inferiority of inactivated vaccines with a similar safety profile. Currently, it is thought that a successful immune response to MTB is defined by a CD4+ T-cell response, supported by CD8+ T-cells and Th17 cells. Thus, these compartments should be activated by a vaccine [115]. Another theory how vaccines such as BCG might work is the concept of “trained innate immunity” through epigenetic changes [116]. Deduced from the above-elicited pathways one might hypothesize that RNA sensing might promote these nonspecific effects for trained immunity. More specifically, enhanced endosomal RNA recognition could promote CD4+ T-cell differentiation, predominantly the Th1/Th17 compartment, as well as CD8+ T-cell activation. Cytosolic RNA sensing could further enhance this response through type I IFNs in the early phase (Figure 3). This would altogether mean a more effective adaptive immune response, therefore eventually impacting the memory response, as well. In the following, the evidence for these hypotheses is presented.

### 4.1. Using Potential Effects of Endosomal RNA Sensing to Generate New Vaccines

In a model using murine embryonic derived, self-renewing alveolar lung macrophages, a dramatically enhanced and earlier cytokine production after infection with vital in comparison to heat-killed MTB bacteria was observed for TNFα, IL-1α, IL-1β, and IL-10 [117]. Adding prokaryotic mRNA as an adjuvant to a vaccine with dead bacteria can greatly amplify the immune response and promote IG-class switching in a mouse model reaching a similar response as live attenuated vaccines. This indicates that RNA sensing plays a role in the observed difference of the immune response to vaccination-induced by viable versus dead bacteria [50]. This observation has been attributed to an enhanced TfH response both in mice [51] and humans [8]. As discussed above, the role of TfH cells to promote humoral response is not the focus for TB immunity. Nevertheless, IL-21, the key cytokine of TfH cells induced by RNA sensing, has been suggested to correlate with vaccine efficacy in mice challenged with mycobacteria [118]. In line with this, adding IL-21 as an adjuvant has increased immune response to a vaccine for *Toxoplasma gondii* in mice [119].

In order to improve the BCG vaccine mice were vaccinated with a recombinant BCG (rBCG) strain expressing IL-12p70 (which would be induced by endosomal RNA sensing, see Figure 3, pathway a) and ESAT-6 leading to an increased Th1 response [120]. However, this did not correlate with protection, addressing the problem that a good marker of vaccination efficacy for TB currently is still lacking. These days it is believed that central memory T cells (TCM) might be more important than effector memory T cells (TEM) in successful vaccination. Adding the human TLR8 agonist TL8-506 (and thus mimicking RNA activation) to a vaccine with ESAT-6 and alum led to a sustained generation of CD8+ TCM in transgenic mice [20], which could provide a strong Th1 response upon infection with MTB mainly mediated through type I IFNs. Furthermore, for immunization with ovalbumin, a common T-cell-dependent antigen, and alum, the addition of TL8-506 led to a prolonged subset of IFNγ-secreting cells in the spleen by CD4+ TCM cells [20]. In support of TLR8 possibly playing a role in vaccine immunity, it was reported that in an Indian TB cohort protection from BCG was associated with a hypermorphic TLR8 SNP [8].

BCG is used in neonates who might lack the capacity of a strong Th1 answer due to their still very immature immune system. This might influence BCG success. Experiments with CLO75, another TLR8 agonist, in monocyte-derived (mo)DCs from newborns showed that CLO75 delivered as encapsulated nanoparticles induced similar levels of IL-1β, IL-6, IL-10 compared to BCG and even higher levels of IL-12p70, and TNFα in vitro [121]. Thus, TRL8 activation might help improve the immune response of newborns to early live attenuated vaccines that require a strong Th1 response. In a humanized mouse model, a combination of CLO75 polymersome nanoparticles with the mycobacterial AG85B induced similar levels of antigen-specific CD4+ T-cells as BCG. Similar experiments have been carried out with poly-I:C, an agonist of TLR3, whose delivery by nanoparticles in combination with BCG infection led to synergistic production of NO in murine BMDM [68].

### 4.2. Canonical BCG and Its Potential Improvement through Cytosolic RNA Sensing

The key difference between MTB and BCG is that BCG lacks the region of deletion (RD)-1 in the genome, which encodes for the proteins ESX, ESAT-6, and culture filtrate protein (CFP)-10. These are necessary for translocation of antigens and eventually the whole bacterium from the infected phagosomes to the cytosol. Thus, BCG does not translocate to the cytosol and cytosolic antigen presentation does not take place. As a result, virulence between MTB and BCG differs significantly, as well as the triggered immunological pathways and memory response. With regard to RNA sensing, cytosolic activation of RLRs, PKR, and NLRP3 will thus not happen, but these receptors were shown to be able to contribute to mycobacterial growth restriction [60]. It was shown that an rBCG strain, which, in contrast to canonical BCG, has access to cytosolic presentation of antigens, leads to superior protection against TB correlating with higher levels of central TfH memory cells in mice [122] (Figure 3, pathway b). However, adding ESX-1 to BCG, and thereby promoting cytoplasmatic antigen presentation, also increased virulence in immunodeficient mice significantly. Thus, attenuated forms of the ESX system have been investigated: The ESX system of *M. marinum* has been introduced to BCG and it could be shown that, in mice, this leads to an increase of MTB-specific Th1 and CD8+ cells by increasing the cGas/STING/TBK1/IRF3/type I interferon axis, as well as by activating the inflammasome via AIM2 and NRLP3 [123]. Though, until now, higher protection evoked by enhanced cytosolic antigen presentation was rather attributed to DNA sensing PRRs, the finding of higher TfH frequency, as well as NLRP3 induction supports the idea of RNA sensing also being involved.

## 5. Conclusions

It has been argued that the very first response to infection with MTB defines its success [124]. Thus, it seems plausible to reexamine PRRs and APCs to find new prophylactic and therapeutic targets. Viable RNA could be easily neglected in vitro experiments but might be a key factor for protective host immunity. Since co-stimulation of several PRRs seems important, endosomal RNA sensing with the triggered cytokines IL-12 and IL-21 might be a crucial signal for mounting a strong Th1 response. As elicited above, IL-21 acts upon multiple cell lines and there is mounting evidence that it plays a key role next to IL-12 in orchestrating the immune response, ultimately both supporting Th1 response by cytotoxic and NK cells, as well as the humoral response. Cytosolic RNA sensing seems to significantly enhance Th1/17 and CD8+ T-cell response through type I IFNs. These pathways might be used in vaccine design to improve the immune response regarding both, trained and adaptive immunity. Evidence for a directly applicable benefit from this new understanding is still weak, but nevertheless promising.

## Figures and Tables

**Figure 1 vaccines-08-00067-f001:**
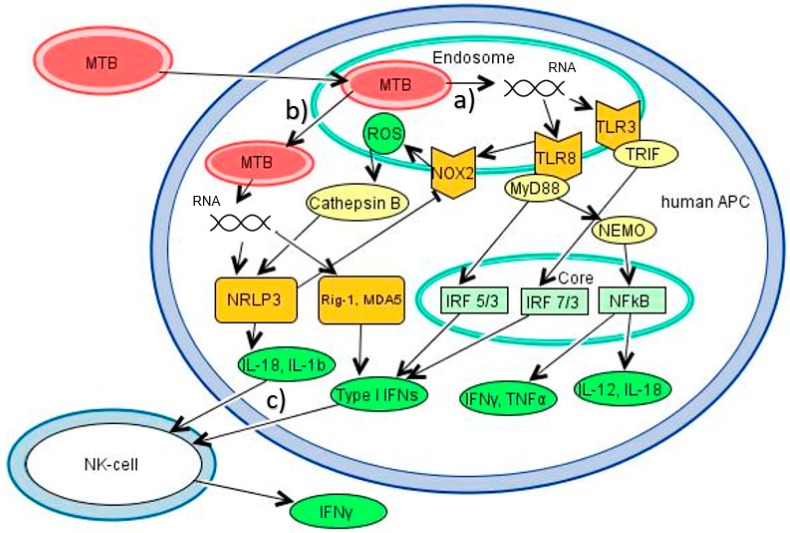
Main immunological pathways in human antigen presenting cells (APCs) after mycobacterial RNA recognition. (**a**) Mycobacteria are engulfed into phagosomes of macrophages and APCs, where RNA is released, eventually stimulating the endosomal receptors TLR8 (ssRNA) and -3 (dsRNA). Activated TLR8 with its adaptor protein MyD88 induces a signal transduction cascade including NF-kappa-B essential modulator (NEMO) ultimately leading to the translocation of NF-κB into the nucleus. Furthermore, mainly interferon response factor (IRF)-5 and to a minor part IRF-7 translocate into the nucleus and activate genes. TLR3 with the adapter protein TRIF leads to translocation of IRF-3 and -7 into the nucleus. NFκB promotes the synthesis of IL-18, IL-12, IFNγ and TNFα, while IRFs promote the production of type I interferons, which activate natural killer (NK) cells to further promote IFNγ in the sense of a feed-forward loop [23]. At the same time, activation of TLR8 enhances production of reactive oxygen species (ROS) through activation of Cytochrome b (-245) beta (CYBB)/NADPH oxidase 2 (NOX2), which will themselves activate the canonical inflammation pathway through Cathepsin B translocating to the cytosol [52,73]. (**b**) After transition to the cytosol, mycobacterial RNA stimulates NLRP3 of the inflammasome, which lead to caspase-1 dependent production of IL-18, IL-1β and pyroptosis [50,51], as well as the RLRs Rig-1 and MDA5, leading to expression of type I IFNs [60]. These cytokines will again stimulate NK cells to produce IFNγ. (**c**) Simultaneously, activation of NLRP3 inhibits CYBB/NOX2 in the sense of a negative-feedback loop [75].

**Figure 2 vaccines-08-00067-f002:**
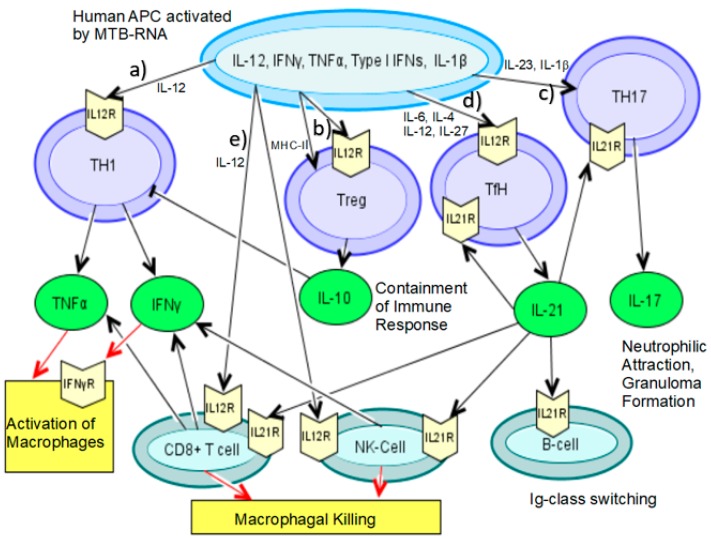
Induction of T-cell compartments after activation of human APCs with mycobacterial RNA. (**a**) At the beginning of the cascade triggered by human APC activation through mycobacterial RNA is IL-12, which will induce naive T-cells to differentiate into Th1 cells through the active from of IL-12p70 leading to production of IFNγ and TNFα, activating macrophages [23]. (**b**) Th17 T-cells are activated by IL-1β, IL-23, which contains the IL-12p40 unit, and IL-21, leading to granuloma formation through neutrophilic attraction [80]. (**c**) Furthermore, IL-12p70 will trigger follicular helper T (TfH) cell differentiation along with IL-6, IL-4, and IL-27 [8]. TfH cells will produce IL-21 interacting with B cells to promote Ig-class switch within the humoral response. (**d**) Simultaneously, both IL-12 and IL-21 will activate NK cells and cytotoxic CD8+ T-cells, which will themselves produce more IFNγ, TNFα, and enhance killing by macrophages [92,104]. (**e**) IL-12p70 also induces regulatory T-cell differentiation along with direct interaction of APCs and naive T-cells through MHC-II molecules. Regulatory T-cells will simultaneously contain the immune-response through IL-10 [17].

**Figure 3 vaccines-08-00067-f003:**
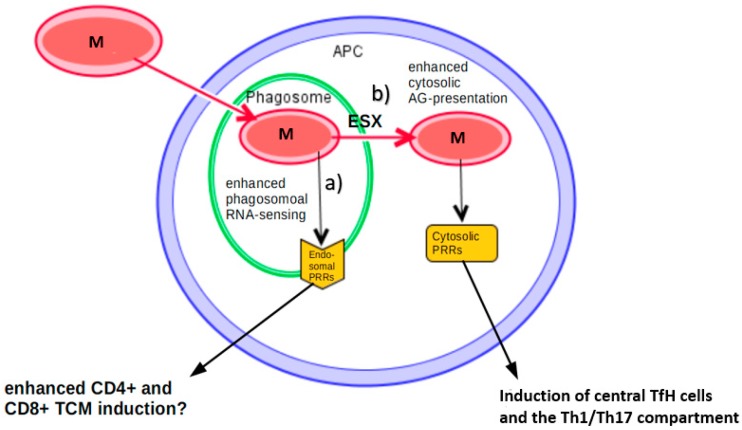
Approaches to improve vaccination against *mycobacterium tuberculosis* (MTB). After intake of a *mycobacterium* (M) into human phagosomes, (**a**) one approach is to enhance signaling from endosomal RNA receptors, which eventually might lead to an enhanced induction of CD4+ and CD8+ T-cells [20,120]. (**b**) Another approach could be to enhance antigens in the cytosol to trigger a stronger immune response by cytosolic RNA receptors, potentially leading to an increased induction of central TfH memory cells, as well as the Th1/Th17 compartment [60,122,123].

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
