# Peer review of "RNA Sensing of Mycobacterium tuberculosis and Its Impact on TB Vaccination Strategies"

_vaccines, 2020, doi:10.3390/vaccines8010067_

Round 1

Reviewer 1 Report

This was a very comprehensive review. At times when reading the review I was unsure whether I was reading a review on TB immunity rather than the role of RNA-sensing in TB vaccination. By including so much detail, the interesting theme of the review was lost.

Though the authors have put in a great deal of effort in this review, I feel that for this review should be revised in the following way;

The early parts of the review should be shortened and should be more focused on how innate immune responses are relevant to RNA-sensing in vaccination against TB. Though this has been done in parts, I feel that there is too much detail regarding TB immunity that may not be relevant to the core theme of the review. The review reads like a list of innate immune responses. The authors could be a bit more creative in their subheadings. For example instead of '3.2 IL-21' how about "IL-21 a key cytokine induced by RNA sensing..." Section 4 requires complete re-writing (see attached). The figures are either poorly referenced within the text and the figure legends do not reference the relevant papers when making specific points. I have attached the manuscript with highlighted suggested changes or areas that need addressing.

Other minor points include;

The use of English is poor in parts or rather unusual. Section 4 requires careful re-writing as this was perhaps the section that exhibited more extensively the poor use of English. There are many sentences and paragraphs that are too long. Some abbreviations are missing e.g. TfH and the explanation should be given first then the abbreviation. A lot of subheadings are vague such as "A word about plasticity'- plasticity of what? In the author contributions what is meant by "visualisation"? The authors mention MSMD, especially in figure 1 but the relevance of the disease to the core theme and its relationship to TB in general are not clearly highlighted. MSMD is only mentioned early on and it is confusing whether the text is in reference to Mtb in general or the MSMD.

Author Response

Point-by-point-reply

For each referee the paragraph of comments was split into single points in order to display how we reacted to every sentence.

Reviewer 1

The early parts of the review should be shortened and should be more focused on how innate immune responses are relevant to RNA-sensing in vaccination against TB. Though this has been done in parts, I feel that there is too much detail regarding TB immunity that may not be relevant to the core theme of the review. The review reads like a list of innate immune responses.

We agree that the first version of our review was too detailed and not focused enough. We shortened paragraphs and deleted details that were not directly related to RNA sensing throughout the manuscript. We believe that the paper now is much more focused and not like a list of innate immune responses. With these changes we also followed similar recommendations of referee 2.

The authors could be a bit more creative in their subheadings. For example instead of '3.2 IL-21' how about "IL-21 a key cytokine induced by RNA sensing..."

We agree with this referee that we were too short with our headings and changed them throughout the manuscript. These changes in our opinion make the content better readable and we introduced more subheadings as suggested by this referee. We agree that this is very helpful for the reader.

Section 4 requires complete re-writing (see attached).

We have restructured this section completely as suggested by this referee and also answered all his points marked within the manuscript. In the new version we agree that the potential impact of RNA sensing on vaccine design becomes much easier to understand.

The figures are either poorly referenced within the text and the figure legends do not reference the relevant papers when making specific points.

We agree that the three figures with legends were not sufficiently integrated into the manuscript in its old version. We added the relevant references to the figure legends and referenced the figures more distinctively within the text. To this end, we added sub-headings into the figures (a, b, c,…), which we termed “pathways”. These subheadings were referred to within the text, which in our opinion helps the reader greatly to illustrate the complex pathways described in the text. With these changes we also followed similar recommendations of referee 2. We thank both reviewers for these very helpful points of critique.

I have attached the manuscript with highlighted suggested changes or areas that need addressing.

We thank the referee for his/her meticulous correction work and agree with all the points. We addressed the highlighted areas and answered/changed all points accordingly. This can easily be followed through the word-correction markup mode.

The use of English is poor in parts or rather unusual. Section 4 requires careful re-writing as this was perhaps the section that exhibited more extensively the poor use of English. There are many sentences and paragraphs that are too long.

We checked the entire manuscript for correct use of the English language and improved many sentences. Particularly we changed the language within the highlighted parts, and we completely re-wrote section 4 following the earlier suggestion and the critique of referee 2. Very long sentences and paragraphs have been shortened throughout the entire manuscript.

Some abbreviations are missing e.g. TfH and the explanation should be given first then the abbreviation. A lot of subheadings are vague such as "A word about plasticity'- plasticity of what?

We carefully re-checked all the used abbreviations and ensured that upon first used they are explained within the text. Answering an earlier point we rewrote numerous subheadings. The paragraph about plasticity was deleted in order to make this review more focused.

In the author contributions what is meant by "visualisation"?

This sentence was omitted.

The authors mention MSMD, especially in figure 1 but the relevance of the disease to the core theme and its relationship to TB in general are not clearly highlighted. MSMD is only mentioned early on and it is confusing whether the text is in reference to Mtb in general or the MSMD.

We agree that the way MSMD was highlighted in the figures did not relate to the text. Thus, we deleted it, and are mentioning it only in line 320-324 in order to argue for the importance of the key cytokines induced by RNA in TB immunity: IL-12 and IFNg.

Reviewer 2 Report

This is a really comprehensive, sometimes sprawling review that goes somewhat beyond just 'RNA-Sensing of Mycobacterium tuberculosis and its impact on TB vaccination strategies.'

It covers a number of interesting points.

Because of the enormous breadth of this piece, there is sometimes a loss of focus which might make it hard for some readers to digest.

I found the Introduction somewhat curious: the topic was introduced over the first few pages from a very high-level, seemingly aimed at a readership who might be unaware what Mtb is. Surely for this readership one might start the story further along, with innate sensing?

Why is 'Tuberculosis' written with a capital T?

The notion that Mtb interacts with innate and adaptive immunity in a way that 'differs from other bacteria' would be misleading to many readers: sure each bacterial infection is a little different in the nuance of its interactions, but the implication here is that there is something unique.

Figure 1 is useful but I would have been aided by specific citations in the legend.

On page 8 the review switches gear entirely onto the subject of adaptive immunity. Given the title of this piece, one might have hoped for a better conceptual link-up between the earlier pages on innate recognition and then this section on the flavour of adaptive response - that is, the specifics of how one shapes the other.

Page 8 also has some rather specific comments about mouse LN experiments: what generalisation is implied, and what is the citation?

Section 4 on implications for vaccination didn't really do justice to the enormous diversity of approaches currently in trials, so felt superficial.

Author Response

Reviewer 2

Because of the enormous breadth of this piece, there is sometimes a loss of focus which might make it hard for some readers to digest.

In answering this similar point 1 of referee 1 we shortened the review in many points and focus now on findings directly linked to mycobacterial RNA-sensing. Thus, the review has become much shorter and in our opinion more easily to read. Furthermore, we restructured the paragraphs to make it more comprehensible.

I found the introduction somewhat curious: the topic was introduced over the first few pages from a very high-level, seemingly aimed at a readership who might be unaware what Mtb is. Surely for this readership one might start the story further along, with innate sensing?

We shortened the introduction substantially and left out the more general points on TB. In consequence we started earlier with the innate immune recognition of Mtb as requested by this referee. We agree that the new version is more suitable for the potential readership of this review.

Why is 'Tuberculosis' written with a capital T?

We apologize for this (“German”) mistake and changed it to a minor “t”.

The notion that Mtb interacts with innate and adaptive immunity in a way that 'differs from other bacteria' would be misleading to many readers: sure each bacterial infection is a little different in the nuance of its interactions, but the implication here is that there is something unique.

We agree that this specific wording in the old version may have been misleading and reworded it (line 290).

Figure 1 is useful but I would have been aided by specific citations in the legend.

We agree that the three figures with legends were not sufficiently integrated into the manuscript in its old version. We added the relevant references to the figure legends and referenced the figures more distinctively within the text. To this end, we added sub-headings into the figures (a, b, c,…), which we termed “pathways”. These subheadings were referred to within the text, which in our opinion helps the reader to illustrate the complex pathways described in the text. With these changes we also followed similar recommendations of referee 1.

On page 8 the review switches gear entirely onto the subject of adaptive immunity. Given the title of this piece, one might have hoped for a better conceptual link-up between the earlier pages on innate recognition and then this section on the flavour of adaptive response that is, the specifics of how one shapes the other.

We thank this referee for this valuable comment and tried to improve on the link-up of adaptive and innate immune system, for example by explicitly explaining the underlying hypothesis (line 427-438). To highlight our important points, we reduced the part about general adaptive immunity, e.g. we deleted the paragraph about plasticity and memory cells.

Page 8 also has some rather specific comments about mouse LN experiments: what generalisation is implied, and what is the citation?

We agree, that these LN experiments distract from the topic and reworded it more generally to make the point of how innate and adaptive immunity are linked.

Section 4 on implications for vaccination didn't really do justice to the enormous diversity of approaches currently in trials, so felt superficial.

We thank this referee for his/her comment and agree completely. However, to include a more complete review of novel TB vaccination strategies would be beyond the scope of this manuscript. We therefore restructured this section completely, focusing on new approaches that relate to RNA-sensing only. Thus, we hope that the impression of superficiality does not apply anymore as we do not claim to give justice to all new approaches of new TB vaccines in the new version of this review.

Round 2

Reviewer 2 Report

This revised manuscript is much improved

Author Response

Thanks for the helpful comments, another check for English language and style has been done.